# Prevalence of stunting and associated factors among under-five children in sub-Saharan Africa: Multilevel ordinal logistic regression analysis modeling

Wullo Sisay Seretew[1]*, Getayeneh Antehunegn Tesema[1], Bantie Getnet Yirsaw[1], Girum Shibeshi Argaw[2]

1 Department of Epidemiology and Biostatistics, Institute of Public Health, College of Medicine and Health Sciences and Comprehensive Specialized Hospital, University of Gondar, Gondar, Ethiopia, 2 Department of Nursing, College of Medicine and Health Science, Jigjiga University, Jigjiga, Ethiopia

* wsisay2733@gmail.com

## Abstract

### Introduction

Stunting is still a major public health problem all over the world, it affecting more than one-third of under-five children in the world that leads to growth retardation, life-threatening complication and accelerate mortality and morbidity. The evidence is scarce on prevalence and associated factors of stunting among under-five children in Sub-Saharan Africa for incorporated intervention. Therefore this study aimed to investigate the prevalence and determinants of stunting among under-five children in Sub-Saharan Africa using recent demographic and health surveys of each country.

### Methods

This study was based on the most recent Demographic and Health Survey data of 36 sub-Saharan African countries. A total of 203,852(weighted sample) under-five children were included in the analysis. The multi-level ordinal logistic regression was fitted to identify determinants of stunting. Parallel line (proportional odds) assumption was cheeked by Brant test and it is satisfied (p-value = 0.68) which is greater than 0.05. Due to the nested nature of the dataset deviance was used model comparison rather than AIC and BIC. Finally the adjusted odds ratio (AOR) with 95% CI was reported identify statistical significant determinants of stunting among under-five children.

### Results

In this study, the prevalence of stunting among under-five children in Sub-Saharan Africa 34.04% (95% CI: 33.83%, 34.24%) with a large difference between specific countries which ranges from 16.14% in Gabon to 56.17% in Burundi. In the multi-level ordinal logistic regression good maternal education, born from mothers aged above 35 years, high household wealth status, small family size, being female child, being female household head, having

**Data Availability Statement:** All the data files are available from the measure DHS program. Data can be accessed from www.measuredhs.com.

**Funding:** The authors received no specific funding for this work.

**Competing interests:** The authors have declared that no competing interests exist.

**Abbreviations:** ANC, Antenatal Care; AOR, Adjusted Odds Ratio; CSA, CI, Confidence Interval; DHS, Demographic and Health Survey; EAs, Enumeration Areas; HAZ, Z-score for Height for-Age; ICC, Intra-cluster Correlation Coefficient; LLR, Log likelihood ratio; SSA, Sub-Saharan Africa; WHO, World Health Organization.

media exposure and having consecutive ANC visit were significantly associated with lower odds of stunting. Whereas, living from rural residence, being 24–59 month children age, single or divorced marital status, higher birth order and having diarrhea in the last two weeks were significantly associated with higher odds of stunting.

## Conclusion

Stunting among under-five children is still public health problem in Sub-Saharan Africa. Therefore designing interventions to address diarrhea and other infectious disease, improving the literacy level of the area and increase the economic level of the family to reduce the prevalence of stunting in the study area.

## Background

Stunting is defined as a height that is more than two standard deviations below the World Health Organization (WHO) child growth standard media. Stunting is still a major public health problem all over the world [1, 2]. World Health organization and the World Bank indicate that the number of stunted children is approximately 150 million, accounting 22% of the children in the world [3]. Furthermore the number of stunting children is concentrated in low-middle- income countries (47%) compared to high-income countries (10%). Around 58.7 million stunted children live in Africa [4], among this majority of the stunted children found in Sub-Sahara Africa regions. Nearly, half of infant and child mortalities in Sub-Sahara Africa are associated with stunting, which culminate in 9% decline in the region's workforce, and thus, hampering the economic growth of the nation [5–7].

Globally, about 144 million under-five children are suffering from stunting, of which 92% are in Low and lower-middle-income countries and Sub-Saharan countries share just over a third of the burden [7]. Considering the adverse effects of malnutrition, particularly on the most vulnerable groups, such as children below 5 years of age, each of the Sub-Saharan Africa countries government initiated and implemented a range of nutritional programs and policies to effectively reduce child malnutrition, whose core targets include a substantial reduction of childhood under-nutrition [6]. Despite all these efforts, there has been little or no reduction in stunting among children below 5 years of age in Sub-Saharan Africa. Therefore, designing effective nutritional intervention policies targeting children at heightened odds of chronic under nutrition is crucial and entails a clear delineation of factors associated with stunted children in the region. Stunting can be caused by various factors such as parental, socio-demographic, and economic status, as well as cultural practices and environmental and other health related variables. For instance, poverty, low parental education, lack of sanitation, low food intake, poor feeding practices, inadequate breastfeeding, repeated infections, family size and birth interval are regarded as key determinants of stunting [8, 9].

To the best of our searching, there is limited evidence regarding stunting prevalence and associated factors among under-five children in the study area in order to improve child nutrition in Sub-Saharan Africa, an individual level and community level based study with sufficient sample size is needed to provide a comprehensive understanding of the factors associated with level of stunting. Hence, the main aim of this study is to utilize data from 36 Sub-Saharan Africa (12 East Africa country, 13 West Africa country, 7 Central Africa and 4 South Africa country) Demographic and Health Survey to determine the factors that are statistical significantly associated with stunting among under-five children after controlling for potential

confounding factors. Findings from this study can be useful to policy makers, program developers, global health sector higher officials and public health researchers in formulating effective interventions and to reach a deliberate equity-driven policy targeting interventions for the most vulnerable population as early as possible.

## Methods

### Study design and setting

Data analysis was done based on the most recent Demographic Health Survey datasets conducted in 36 sub-Saharan African countries. DHS is a nationally representative survey that provides data for monitoring indicators of nutrition and health. The authorization to use the data was approved from the measure DHS program.

### Study population and sampling procedure

All under-five children in Sub-Saharan Africa within the five years preceding the recent survey of each Sub-Saharan Africa countries were the source population and All under-five children in Sub-Saharan Africa who were in the selected clusters were considered as the study population, Children in the study who were in the selected clusters that complete data all information of the interesting variable were included. Children with incomplete information of the interesting variables were excluded from the analysis. The nature of DHS, some of the regions or counties were oversampled and some were under sampled in particular in large counties or regions. The data were weighted before any statistical analysis to restore the representativeness of the data and to get a reliable estimate and standard error. A total weighted sample of 203,852 under five children was including. A multistage cluster sampling technique was employed to recruit the samples using Enumeration areas as primary sampling units and households as secondary sampling units.

### Variables of the study

The outcome variable for our study was the stunting status of children, which was an ordered category variables categorized in to three ordinal categories. Those categories was normal, moderate stunting (height-for-age Z-score (HAZ) $<-2SD$) and sever stunting (height-for-age Z-score (HAZ) $<-3SD$).

The explanatory variables were classified in to individual-and community-level variables. Type of residence, Child age, child sex, household wealth, maternal education, maternal age, sex of household head, child twin, birth order, media exposure, number of ANC visits, place of delivery, husband education, birth size, and preceding birth interval were individual-level variables. Residence and sub-Saharan African region were community-level variable.

### Data management and analysis

STATA version 17 software was used for data management and analysis. Descriptive statistics were done using frequencies and percentages. For ordered nature of the response variable and the hierarchal nature of the DHS data violates the independent assumptions of the standard binary logistic regression model and because of this, we cannot use the standard logistic regression analysis rather than we can fitted a multilevel ordinal logistic regression analysis. They are different types of ordinal logistic regression models. The most common logistic regression models are: Proportional Odds Model, Partial Proportional Odds Model with Restrictions and Partial Proportional Odds Model without Restrictions, Continuous Ratio Model and Stereotype Model. In most studies of the glove use Proportional Odds Model is the most commonly

used ordinal logistic regression model. The fundamental assumption in an ordinal logistic regression model is the Parallel line assumption. If the data satisfies the Parallel line assumption, the proportional odds model can be used.

We have checked the parallel line assumptions, which state that the effects of all independent variables are constant across categories of the response variable. The Brant test discovered that the parallel line assumption was fulfilled (p>0.05). In addition, the DHS data has a hierarchical nature. For that reason, children were nested within a cluster, and we assume that study subjects in the same cluster may distribute similar characteristics to participants in another cluster. This implies the want to take into account the heterogeneity between clusters by using an advanced model. Then, a multilevel proportional odds model was performed. As a result, because the Brant test was met, the multilevel proportional odds model gave a single Odds Ratio (OR) for an explanatory variable (severe vs. moderate/ normal, and severe/moderate vs. normal).

Whereas conducting a multilevel ordinal logistic regression analysis four models; the null model containing only the response variable, model I and II containing individual and community level variables, respectively and model III, which contains both individual and community level variables were fitted. Interclass Correlation Coefficient (ICC), Median Odds Ratio (MOR) and Percentage Change in Variance (PCV) were checked to assess the clustering effect. Since these models were nested, we used deviance to check the model comparison and the model with the lowest deviance were chosen as the best model for the data. Both bi-variable and multivariable multilevel ordinal logistic regression was done and variables with a p-value of <0.2 in the bi-variable multilevel ordinal logistics analysis were considered for multivariable multilevel ordinal logistics analysis. Finally, variables with P-value <0.05 in the multivariable multilevel ordinal logistics analysis were identified as significant factors associated with stunting.

### Ethical approval and consent to participate

Ethical approval for this study was not necessary since this study used existing public domain survey data sets. We requested the data from the MEASURE DHS Program and permission was granted to download and use the data for this study. There are no names of individuals or household addresses in the data files.

## Results

### Socio-demographic characteristic of study participants

The prevalence of stunting among under-five children in Sub-Saharan Africa 34.04% (95% CI: 33.83%, 34.24%) with a large difference between specific countries which ranges from 16.14% in Gabon to 56.17% in Burundi Table 1.

Among a total of 203,852 study participants, of this study participants approximately one-thirds (38.96%) of the mothers did not have formal education. Nearly half of (53.34%) of the children's mothers were aged 25–35 years. Around two-third of the (64.06%) children's mothers were at list one ANC follow up during her pregnancy time. Most of the study participants (96.9%) were single birth types. Regarding place of delivery around two-thirds of (66.39%) the children got birth at a health facility and 56.76% participants had 1–3 birth order history. About 69.42% of the study participants lived in rural areas.

We looking the wealth status of the study participants 35.67%, 20.08%, 44.25% were rich, middle and poor respectively. About 170,892 (83.87%) had no diarrhea and fever in the last two weeks. Male (50.32%) and female (49.68%) children were almost equally represented. The majority of the study participants (72.75%) were married marital status. Around three-fourth

**Table 1. Number of study participants, prevalence of stunting and year of the survey for each country of our study.**

| Sub-Saharan Africa region | | Sample Size (n = 203 852) | Prevalence of stunting | Year of the survey |
|---|---|---|---|---|
| Central Africa | Angola | 5905 | 37.42 | 2015/2016 |
| | Chad | 9805 | 39.75 | 2014/2015 |
| | Congo | 3859 | 23.45 | 2011/2012 |
| | DR. Congo | 7900 | 42.56 | 2013/2014 |
| | Cameron | 5053 | 32.32 | 2011 |
| | Gabon | 2792 | 16.14 | 2012 |
| | São Tomé's Príncipe | 1284 | 29.63 | 2008/2009 |
| East Africa | Burundi | 6222 | 56.17 | 2016/2017 |
| | Comoros | 2468 | 29.73 | 2018 |
| | Ethiopia | 9588 | 38.70 | 2016 |
| | Kenya | 17 291 | 23.01 | 2014 |
| | Madagascar | 4994 | 50.55 | 2021 |
| | Malawi | 5159 | 36.82 | 2015/2016 |
| | Mozambique | 9895 | 43.16 | 2011 |
| | Rwanda | 3593 | 38.13 | 2019/2020 |
| | Tanzania | 8814 | 34.45 | 2015/2016 |
| | Uganda | 4350 | 28.38 | 2016 |
| | Zambia | 8612 | 34.69 | 2018 |
| | Zimbabwe | 5212 | 26.65 | 2013/2014 |
| **South Africa** | Lesotho | 1298 | 32.85 | 2014 |
| | Namibia | 1438 | 22.33 | 2013 |
| | Swaziland | 2070 | 28.00 | 2006/2007 |
| | South Africa | 1080 | 26.55 | 2016 |
| **West Africa** | Burkina Faso | 6660 | 34.67 | 2010 |
| | Benin | 11 717 | 31.84 | 2017/2018 |
| | Coˆte d'Ivoire | 3041 | 29.97 | 2011 |
| | Ghana | 2636 | 18.20 | 2014 |
| | Gambia | 2939 | 24.33 | 2019/2020 |
| | Guinea | 3374 | 31.41 | 2018 |
| | Mali | 8858 | 26.94 | 2018 |
| | Nigeria | 11 407 | 36.70 | 2018 |
| | Niger | 5113 | 43.54 | 2012 |
| | Liberia | 2754 | 30.36 | 2019/2020 |
| | Sierra leone | 4141 | 37.98 | 2019 |
| | Senegal | 3357 | 26.64 | 2010/2011 |
| | Togo | 9173 | 26.95 | 2013/2014 |

of (79.36%) the household heads were males, and the majority (94.92%) of the study participants were Delivery by without caesarean section Table 2.

## Prevalence of stunting in sub-Saharan Africa

The overall prevalence of stunting among under-five children was 34.04% [95% CI: 33.83%, 34.24%]. Our study showed that 20.13% [95% CI: 19.95%, 20.30%] of under-five children had moderate stunting and 13.91% [95% CI: 13.76%, 14.06%] severe stunting. The highest prevalence of stunting was found in children whose mothers were not ANC visit, place of delivery at home and poor wealth status which was 41.32%, 40.76% and 40.54%, respectively. Regarding

**Table 2. Descriptive characteristics of the study participants in Sub-Saharan Africa.**

| Variable | Weighted Frequency(N) | Weighted frequency (%) |
|---|---|---|
| **Maternal education level** | | |
| No education | 79432 | 38.96 |
| Primary | 72871 | 35.75 |
| Secondary and above | 51549 | 25.29 |
| **Maternal age** | | |
| Below 25 | 56907 | 27.92 |
| 25–35 | 108739 | 53.34 |
| Above 35 | 38206 | 18.74 |
| **Birth type** | | |
| Single | 197 541 | 96.90 |
| Twin | 6311 | 3.10 |
| **Birth order** | | |
| 1–3 | 115707 | 56.76 |
| 4–6 | 61740 | 30.29 |
| >6 | 26405 | 12.95 |
| **Place of delivery** | | |
| Health facility | 135342 | 66.39 |
| Home | 68510 | 33.61 |
| **ANC visit** | | |
| No | 73271 | 35.94 |
| Yes | 130581 | 64.06 |
| **Place of residence** | | |
| Urban | 62329 | 30.58 |
| Rural | 141523 | 69.42 |
| **Sex of household head** | | |
| Male | 161784 | 79.36 |
| Female | 42068 | 20.64 |
| **Wealth status** | | |
| Poor | 90211 | 44.25 |
| Middle | 40922 | 20.08 |
| Rich | 72719 | 35.67 |
| **Family size** | | |
| <6 | 83837 | 41.12 |
| 6–9 | 85816 | 42.10 |
| >9 | 34199 | 16.78 |
| **Source of drink water** | | |
| Protected | 107747 | 52.86 |
| Unprotected | 96076 | 47.14 |
| **Marital status** | | |
| Married | 148295 | 72.75 |
| Single/divorced/separated | 55557 | 27.25 |
| **Sex of a child** | | |
| Male | 102582 | 50.32 |
| Female | 101270 | 49.68 |
| **Child's age** | | |
| <6 months | 25640 | 12.58 |
| 6–23 months | 61744 | 30.29 |
| 24–59 months | 116468 | 57.13 |

(*Continued*)

**Table 2.** (Continued)

| Variable | Weighted Frequency(N) | Weighted frequency (%) |
|---|---|---|
| **History of diarrhea two weeks prior to the survey** | | |
| No | 170892 | 83.87 |
| Yes | 32867 | 16.13 |
| **History of intake of parasite drugs for mothers during pregnancy** | | |
| Yes | 62236 | 45.82 |
| No | 73586 | 54.18 |
| **Work status** | | |
| Yes | 120635 | 59.18 |
| No | 83217 | 40.82 |
| **Media exposure** | | |
| Yes | 134568 | 66.01 |
| No | 69284 | 33.99 |
| **Husband education level** | | |
| No education | 61109 | 34.16 |
| Primary | 101901 | 56.96 |
| Secondary and above | 15893 | 8.88 |
| **Delivery by caesarean section** | | |
| Yes | 10276 | 5.08 |
| No | 192190 | 94.92 |
| **Size of child at birth** | | |
| Small | 68740 | 35.30 |
| Normal | 92415 | 47.46 |
| Large | 33581 | 17.24 |

the severity of stunting, the highest prevalence of severe stunting was found in children whose family had no media exposure (19.91%) and children got from illiterate fathers (18.45%) and the highest prevalence of moderate stunting were found small size of child at birth (23.20%) and the age of the child 24–59 months(22.66%) Table 3.

## Determinants of stunting among under-five children

To identify the determinants of stunting, the bi-variable analysis was conducted and a variable p-value <0.02 includes in the multivariable analysis. Both the individual and country-level factors were included simultaneously, sex of household head, sex of a child, media exposure, ANC visit, place of delivery, birth order, maternal education level, wealth status of the family, source of drink water, father education level, size of a child at birth, age of the child, maternal age and place of residence were statistically significant with stunting.

Children who were the 4th-6th and greater than 6th birth order were 1.11 times (AOR = 1.11, 95% CI: 1.07, 1.15) and 1.23 times (AOR = 1.23, 95% CI: 1.17, 1.29) higher odds of stunting compared to 3rd and below birth order respectively. Female headed households had 0.97(AOR = 0.97, 95%CI: 0.94–0.99) times lower odds of stunting compared with male headed households.

Looking at educational status of mother, mother with primary (AOR = 0.95, 95%CI: 0.92–0.98) and secondary and above educational status (AOR = 0.70, 95%CI: 0.67–0.73) had lower odds of stunting compared with illiterate mother. The odds of being at higher stunting among

**Table 3. The prevalence of stunting based on the individual level, community level, child and maternal characteristics in sub-Saharan Africa.**

| Characteristics | Categories | Stunting status | | | Over all stunting pr for each categories |
|---|---|---|---|---|---|
| | | Sever | Moderate | Normal | |
| Maternal education level | No educated | 18.32 | 21.49 | 60.19 | 39.81 |
| | Primary | 13.98 | 21.87 | 64.15 | 35.85 |
| | Secondary and above | 7.02 | 15.56 | 77.42 | 22.58 |
| Maternal age | Below 25 | 13.94 | 20.97 | 65.09 | 34.91 |
| | Between 25& 35 | 13.70 | 19.60 | 66.70 | 33.30 |
| | Above 35 | 14.46 | 20.34 | 65.20 | 34.80 |
| Birth type | Single | 13.64 | 19.93 | 66.43 | 33.57 |
| | Twin | 22.43 | 20.13 | 65.96 | 34.04 |
| Birth order | 1–3 | 12.54 | 19.53 | 67.93 | 32.07 |
| | 4–6 | 15.07 | 20.73 | 64.19 | 35.81 |
| | >6 | 17.19 | 21.31 | 61.51 | 38.49 |
| Place of delivery | Health facility | 11.43 | 19.21 | 69.36 | 30.64 |
| | Home | 18.80 | 21.94 | 59.26 | 40.76 |
| ANC visit | Yes | 11.57 | 18.94 | 69.49 | 30.51 |
| | No | 18.07 | 22.24 | 59.68 | 41.32 |
| Place of residence | Urban | 8.50 | 16.20 | 75.30 | 24.70 |
| | Rural | 13.91 | 20.13 | 65.96 | 34.04 |
| Sex of household head | Male | 14.15 | 20.09 | 65.76 | 34.24 |
| | Female | 13.00 | 20.27 | 66.73 | 33.27 |
| Wealth status | Poor | 17.80 | 22.74 | 59.46 | 40.54 |
| | Middle | 13.97 | 21.28 | 64.75 | 35.25 |
| | Rich | 9.05 | 16.23 | 74.72 | 25.28 |
| Family size | <6 | 12.90 | 19.97 | 67.13 | 32.87 |
| | 6–9 | 14.54 | 20.39 | 65.07 | 34.93 |
| | >9 | 14.81 | 19.85 | 65.34 | 34.66 |
| Source of drink water | Protected | 12.08 | 18.89 | 69.03 | 30.97 |
| | Unprotected | 15.96 | 21.52 | 62.52 | 37.48 |
| Marital status | Married | 14.18 | 20.10 | 65.72 | 34.28 |
| | Single/divorced/separated/widowed | 13.19 | 20.20 | 66.61 | 33.39 |
| Sex of a child | Male | 15.38 | 21.00 | 63.62 | 36.38 |
| | Female | 12.42 | 19.23 | 68.34 | 31.66 |
| Child's age | <6 months | 6.13 | 9.77 | 84.11 | 15.89 |
| | 6–23 months | 12.57 | 19.64 | 67.79 | 32.21 |
| | 24–59 months | 16.33 | 22.66 | 61.01 | 38.99 |
| History of diarrhea two weeks prior to the survey | Yes | 13.55 | 19.95 | 66.50 | 33.50 |
| | No | 15.77 | 21.06 | 63.17 | 36.83 |
| History of intake of parasite drugs for mothers during pregnancy | Yes | 11.23 | 18.89 | 69.88 | 30.12 |
| | No | 14.19 | 19.48 | 66.33 | 33.67 |
| Work status | Yes | 14.08 | 20.66 | 65.26 | 34.74 |
| | No | 13.66 | 19.35 | 66.99 | 33.01 |
| Media exposure | Yes | 11.33 | 18.90 | 69.77 | 30.23 |
| | No | 18.91 | 22.50 | 58.58 | 41.42 |
| Husband education level | No education | 18.45 | 21.38 | 60.17 | 39.87 |
| | Primary | 15.24 | 22.19 | 62.57 | 37.43 |
| | Secondary & above | 9.73 | 17.20 | 73.07 | 26.93 |
| Delivery by caesarean section | Yes | 8.85 | 15.86 | 75.29 | 24.71 |
| | No | 14.14 | 20.36 | 65.50 | 34.50 |

*(Continued)*

**Table 3.** (Continued)

| Characteristics | Categories | Stunting status | | | Over all stunting pr for each categories |
|---|---|---|---|---|---|
| | | Sever | Moderate | Normal | |
| Size of child at birth | Small | 18.80 | 23.20 | 58.00 | 42.00 |
| | Normal | 13.74 | 20.66 | 65.60 | 34.40 |
| | Large | 12.52 | 18.19 | 69.29 | 30.71 |

children who took drugs for the intestinal parasite in the last six months were decreased by 3% (AOR = 0.97, 95% CI: 0.95, 0.99) than those who did not take drugs. Children who had diarrhea in the last two weeks had 1.11times (AOR = 1.11, 95% CI: 1.08, 1.15) higher odds of a higher level of stunting compared to children who did not have diarrhea Table 4.

## Proposed model comparison

A model with low (195,541.62) deviance and high log-likelihood (-97770.82) was the best model. The combined individual level and community level multilevel ordinal logistic regression model (model III) was the best-fitted model in our study.

## Discussion

Our study shows that the extent of stunting among under-five children in Sub-Saharan Africa is high. This might be related to the limited health service infrastructure coverage [4], highly prevalence of infectious disease areas like hookworm, malaria and poor sanitation system of the environment [10, 11] all these factors might be a comfort zone of spread of parasites and their transmissions [12].

The final model of our study indicates that children born to illiterate mother had higher odds of stunting compared to literate mother. It's consistent with studies conducted in Bolivia [13], Bangladesh [14] and Indonesia [8]. It could be due to education can positively influence feeding practice of their children, good understanding regarding with child health and nutrition, encourage to follow antenatal care services during pregnancy and educated mothers are more likely to utilize child health services, which can have a positive effect on their children's health outcomes [15].

Sex of a child was highly associated with stunting in this study, evidence show that male children had higher odds of stunting compared to female children. It is in line with studies conducted in Nigeria, Kenya, Tanzania and India [16–19]. This might be due to male children and female children are different with protein and gene expression placenta especially during adverse conditions in addition to this it might be the mi RNA could show down regulation in male and up regulation in female. This makes males have more stunting than of females. Place of delivery was strong association with stunting. Hence, children who delivered at home had higher odds of stunted as compared with children delivered at health facility. This finding is similar studies conducted in Democratic Republic Congo, Uganda and Iran [20–22] this may be occurred due to lack of adequate new born care at home, the baby may not take the immediate nutrition after birth and its difficult to solve complications.

The result of the current studies showed that children taking drugs for the intestine parasite in the last six months had less odds of stunting compared to children had no taking drugs for the intestine parasite in the last six months it is consistent with previous studies reported in Eastern Nepal, Ethiopia and Tanzania, [11, 23, 24] it could be due to the routine vitamin A supplementation, protective or curative deforming combined with Integrated Management of

**Table 4. Multivariate ordinal logistic regression analysis of stunting against individual level and community level characteristics.**

| Characteristics | Null model | Model I | Model II | Model III |
|---|---|---|---|---|
| | | AOR with 95%CI | AOR with 95%CI | AOR with 95%CI |
| Sex of household head | | | | |
| female | | | 0.96(0.93, 0.99) | 0.97(0.94, 0.99)* |
| Sex of a child | | | | |
| Female | | | 0.75(0.73, 0.76) | 0.75(0.73, 0.76)* |
| Media exposure | | | | |
| yes | | | 0.81(0.79, 0.83) | 0.82(0.80, 0.84)* |
| ANC Visit | | | | |
| Yes | | | 0.81(0.77, 0.84) | 0.80(0.77, 0.84)* |
| Place delivery | | | | |
| Home | | | 1.13(1.10, 1.17) | 1.12(1.09, 1.15)* |
| Birth order | | | | |
| 4–6 order | | | 1.11(1.07, 1.15) | 1.11(1.07, 1.15)* |
| >6 order | | | 1.24(1.18, 1.3) | 1.23(1.17, 1.29)* |
| Twin status | | | | |
| Twin | | | 2.09(1.93, 2.28) | 2.10(1.93, 2.28)* |
| Maternal educationlevel | | | | |
| Primary | | | 0.95(0.92, 0.98) | 0.95(0.92, 0.98)* |
| Secondary and above | | | 0.68(0.66, 0.71) | 0.70(0.67, 0.73)* |
| Wealth status | | | | |
| Middle | | | 0.93(0.90, 0.97) | 0.95(0.92, 0.98)* |
| Rich | | | 0.76(0.73, 0.78) | 0.81(0.79, 0.84)* |
| Family size | | | | |
| 6–9 | | | 0.96(0.93, 0.99) | 0.96(0.93, 0.99)* |
| >9 | | | 0.99(0.96, 1.03) | 0.99(0.96, 1.03) |
| Source of drink water | | | | |
| Protected | | | 0.92(0.90, 0.94) | 0.94(0.92, 0.97)* |
| Marital status | | | | |
| single/divorced/separated | | | 1.06(1.03, 1.09) | 1.07(1.04, 1.10)* |
| Age of child | | | | |
| 6–23 month | | | 2.75(2.64, 2.87) | 2.75(2.64, 2.87)* |
| 24–59 month | | | 4.25(4.07, 4.43) | 4.25(4.07, 4.43)* |
| History of diarrhea two weeks prior to the survey | | | | |
| Yes | | | 1.11(1.08, 1.15) | 1.11(1.08, 1.15)* |
| History of intake of parasite drugs for mothers during pregnancy | | | | |
| Yes | | | 0.97(0.94, 0.99) | 0.97(0.95, 0.99)* |
| Husband education level | | | | |
| Primary | | | 0.82(0.77, 0.88) | 0.83(0.78, 0.88)* |
| Secondary and above | | | 0.72(0.68, 0.76) | 0.74(0.70, 0.78)* |
| Delivery by caesarean section | | | | |
| Yes | | | 0.90(0.84, 0.96) | 0.91(0.85, 0.97)* |
| Size of a child at birth | | | | |
| Normal | | | 1.21(1.17, 1.23) | 1.20(1.17, 1.23)* |
| Low | | | 1.70(1.64, 1.76) | 1.70(1.64, 1.76)* |
| Maternal age | | | | |
| 25–35 | | | 0.83(0.80, 0.86) | 0.83(0.81, 0.86)* |
| Above 35 | | | 0.71(0.68, 0.75) | 0.72(0.68, 0.75)* |

(*Continued*)

**Table 4.** (*Continued*)

| Characteristics | Null model | Model I | Model II | Model III |
|---|---|---|---|---|
| | | AOR with 95%CI | AOR with 95%CI | AOR with 95%CI |
| Residence | | | | |
| Rural | | 1.78(1.74, 1.82) | | 1.21(1.17, 1.25)* |
| cut1 | 2.01(1.98, 2.05) | 1.11(1.08, 1.13) | 1.15(1.08, 1.22) | 1.34(1.26, 1.41) |
| cut2 | 1.87(1.85, 1.89) | 2.29(2.26, 2.31) | 2.38(2.31, 2.45) | 2.57(2.49, 2.65) |
| **Random effect result** | | | | |
| Log-likelihood | -179278.5 | -177790.36 | -97830.07 | -97770.81 |
| Deviance | 358,557 | 355,580.72 | 195,660.14 | 195,541.62 |
| ICC | 0.09(0.07, 0.11) | | | |
| MOR | 1.45(1.38, 1.53) | | | |

AOR = Adjusted Odds Ratio: CI: Confidence Interval: ICC = Intra-class Correlation Coefficient: MOR: Median Odds Ratio:

*P-value < 0.05

Neonatal and Childhood Illness may benefit to decrease the problem of childhood stunting. The current study showed that the odds of stunting was higher among twin pregnancies mother compared with that of singleton pregnancies mother which is in line with study conducted in South Africa [25], East Africa [26] and Italy [27]. This might be multiple pregnancies can lead preterm labor, low birth weight, delivery complication, high risk of birth infection and biological immaturity. Children from large family size had higher stunted as compared with small family size it is consistent with previous studies reported in Nigeria [28], Iran and Indonesia [29, 30] this may be occurred due to children from large family size may not get sufficient nutrients like vitamin B12 and iron.

The study at hand, children born to mother's age less than 25 years higher odds of stunting compared to children born to mother age and above. It is similar with studies reported in Tanzania [6, 9, 18, 31]. The possible reasons might be babies born to aged mother are less likely to be preterm and low birth weight as compared to younger mothers. Our study revealed that children from families with poor household wealth had higher odds of stunting compared to children from rich household wealth. It is in line with study finding in India, Brazil and Kenya [32–34] it may due to poverty is strong association with food insecurity and low high income are more likely to purchase nutrient-rich foods and also the children from high income level might get necessary health services during pregnancy and the child might be provided good care than family with low-income levels. Other significant predictor that was associated with stunting was source of drinking water. Those children whose families use drinking water from protected source were less likely to be stunted as compared to those who used unprotected source. This finding similar with studies conducted in Indonesia, Malawi and Tanzania [11, 12, 35, 36]. This might be because of unsafe water aggravates the spread of water-borne diseases that can affect the health status of children.

Exposure to diarrheal diseases two weeks prior to the survey had significant association with stunting. The finding of this study was in line with the other studies conducted in different parts of the world like USA, Virginia and Peru [37–40]. All the referenced studies confirmed that exposure to diarrhea two weeks prior to the survey had association with stunting. This might be because of loss of fluids and electrolytes, loss of food appetite and absorption in the intestine.

### Strength and limitations of the study

This study has several strengths. First, the study was based on important weighted nationally representative DHS surveys of 36 Sub-Saharan African countries that were to get a reliable estimate. Secondly, multilevel ordinal logistic regression was fitted by considering the hierarchical nature of the DHS data to get identify community and individual-level predictors of stunting among under-five children. Furthermore, the study was based on the large sample size; this could increase the power of the study to get the true effect of the predictors. This finding should be interpreted in light of the following limitations. This study was based upon recall by mothers and it is prone to recall bias and the DHS survey year was not the same in all countries, it was based on DHS conducted 2008 to 2019. This might overestimate or underestimate the prevalence of stunting among under-five children.

## Conclusion and recommendation

The prevalence of stunting among under-five children in Sub-Saharan Africa was high. Our findings indicate that stunting was a major public health problem in Sub-Saharan Africa. Maternal education level, maternal age, birth order, wealth status of family, family size, child age, sex of household head, husband education level, source of drink water, taking drugs for an intestinal parasite, diarrhea in the last two weeks, family size was found statistical significant parameters of stunting in Sub-Saharan Africa region. So, health sector policy makers should focus on improving the education access of the area, give intervention to address diarrhea and other infectious disease, increase the economic level of the family, improving sanitation and water access of the community to reduced the burden of stunting among under-five children for the betterment of the continuity of the generations.

## Supporting information

**S1 Data.**
(DTA)

## Acknowledgments

We would like to thank measure DHS for their permission to access the different Sub-Saharan Africa countries DHS datasets.

## Author Contributions

**Conceptualization:** Wullo Sisay Seretew, Getayeneh Antehunegn Tesema, Bantie Getnet Yirsaw, Girum Shibeshi Argaw.

**Data curation:** Wullo Sisay Seretew, Getayeneh Antehunegn Tesema, Bantie Getnet Yirsaw, Girum Shibeshi Argaw.

**Formal analysis:** Wullo Sisay Seretew, Getayeneh Antehunegn Tesema, Bantie Getnet Yirsaw, Girum Shibeshi Argaw.

**Funding acquisition:** Wullo Sisay Seretew, Getayeneh Antehunegn Tesema, Bantie Getnet Yirsaw, Girum Shibeshi Argaw.

**Investigation:** Wullo Sisay Seretew, Getayeneh Antehunegn Tesema, Bantie Getnet Yirsaw, Girum Shibeshi Argaw.

**Methodology:** Wullo Sisay Seretew, Getayeneh Antehunegn Tesema, Bantie Getnet Yirsaw, Girum Shibeshi Argaw.

**Project administration:** Wullo Sisay Seretew, Getayeneh Antehunegn Tesema, Bantie Getnet Yirsaw, Girum Shibeshi Argaw.

**Resources:** Wullo Sisay Seretew, Getayeneh Antehunegn Tesema, Bantie Getnet Yirsaw, Girum Shibeshi Argaw.

**Software:** Wullo Sisay Seretew, Getayeneh Antehunegn Tesema, Bantie Getnet Yirsaw, Girum Shibeshi Argaw.

**Supervision:** Wullo Sisay Seretew, Getayeneh Antehunegn Tesema, Bantie Getnet Yirsaw, Girum Shibeshi Argaw.

**Validation:** Wullo Sisay Seretew, Getayeneh Antehunegn Tesema, Bantie Getnet Yirsaw, Girum Shibeshi Argaw.

**Visualization:** Wullo Sisay Seretew, Getayeneh Antehunegn Tesema, Bantie Getnet Yirsaw, Girum Shibeshi Argaw.

**Writing – original draft:** Wullo Sisay Seretew, Getayeneh Antehunegn Tesema, Bantie Getnet Yirsaw, Girum Shibeshi Argaw.

**Writing – review & editing:** Wullo Sisay Seretew, Getayeneh Antehunegn Tesema, Bantie Getnet Yirsaw, Girum Shibeshi Argaw.

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
