## [Decision Letter · Decision Letter 0]

5 Sep 2023

PONE-D-23-10610Prevalence of stunting and associated factors among under-five children in sub-Saharan Africa: Multilevel ordinal logistic regression analysis modelingPLOS ONE

Dear Dr. Seretew,

Thank you for submitting your manuscript to PLOS ONE. After careful consideration, we feel that it has merit but does not fully meet PLOS ONE’s publication criteria as it currently stands. Therefore, we invite you to submit a revised version of the manuscript that addresses the points raised during the review process.

We look forward to receiving your revised manuscript.

Kind regards,

Anteneh Mengist Dessie, MPH

Academic Editor

PLOS ONE

2. Thank you for including your ethics statement:  "Ethical approval for this study was not necessary since this study used existing public domain survey data sets. We requested the data from the MEASURE DHS Program and permission was granted to download and use the data for this study. There are no names of individuals or household addresses in the data files.".  

a. For studies reporting research involving human participants, PLOS ONE requires authors to confirm that this specific study was reviewed and approved by an institutional review board (ethics committee) before the study began. Please provide the specific name of the ethics committee/IRB that approved your study, or explain why you did not seek approval in this case.

b. Please provide additional details regarding participant consent. In the ethics statement in the Methods and online submission information, please ensure that you have specified (1) whether consent was informed and (2) what type you obtained (for instance, written or verbal, and if verbal, how it was documented and witnessed). If your study included minors, state whether you obtained consent from parents or guardians. If the need for consent was waived by the ethics committee, please include this information.

Additional Editor Comments:

If the data is a secondary analysis of a survey study that is publicly available, then kindly provide a direct link to the dataset in 'Data Availability'. We noticed that you have already provided a link in 'Describe where the data may be found in full sentences, but that link is a homepage link. Kindly also provide relevant accession numbers or search information in this statement or provide a direct link to the dataset instead of a homepage link. You can also upload a minimal anonymized dataset to the supporting information files.

Reviewers' comments:

Reviewer's Responses to Questions

**Comments to the Author**

1. Is the manuscript technically sound, and do the data support the conclusions?

Reviewer #1: Yes

Reviewer #2: Yes

2. Has the statistical analysis been performed appropriately and rigorously? 

Reviewer #1: Yes

Reviewer #2: Yes

3. Have the authors made all data underlying the findings in their manuscript fully available?

Reviewer #1: Yes

Reviewer #2: Yes

4. Is the manuscript presented in an intelligible fashion and written in standard English?

Reviewer #1: Yes

Reviewer #2: Yes

5. Review Comments to the Author

Reviewer #1: What things did you add for the scientific community or policy makers ? Just beyond using advanced statistical technique or what new thing did you find which is not touched by other literatures??

whereas conducting a multilevel ordinal logistic regression analysis four models; the null model containing only the response variable, model I and II containing individual and community level variables, respectively and model III, which contains both individual and community level variables were fitted. Interclass Correlation Coefficient (ICC), Median Odds Ratio (MOR) and

Percentage Change in Variance (PCV) were checked to assess the clustering effect. Since these models were nested, we used deviance to check the model comparison and the model with the lowest deviance were chosen as the best model for the data.

Why not you usec AIC OR BIC?

Over all, you mansucript is well written but you sholud go through to handle grammatical errors throught your document.

Reviewer #2: Good study. Please elaborate on how the sample size was calculated. Other than that, it is a good, interesting paper. Introduction is precise, methodology is detailed, statistical analysis is appropriate.

6. PLOS authors have the option to publish the peer review history of their article (what does this mean?). If published, this will include your full peer review and any attached files.

Reviewer #1: No

Reviewer #2: No

---

## [Author Response · Author response to Decision Letter 0]

15 Jan 2024

Point-by-point response 

Point-by-point response for Editors/Reviewers' comments 

Scientific Reports

Manuscript title: Prevalence of stunting and associated factors among under-five children in sub-Saharan Africa: Multilevel ordinal logistic regression analysis modeling

Submission ID:PONE-D-23-10610

 Dear editor/reviewer. 

Dear all,

We would like to thank you for the constructive, building, and improvable comments on this manuscript that would improve the content of the manuscript. We considered each comment and clarification question of editors and reviewers on the manuscript thoroughly. Our point-by-point responses for each comment and question are described in detail on the following pages. Further, the details of changes were shown by track changes in the supplementary document attached

'Response to Editors’ comment

1. If the data is a secondary analysis of a survey study that is publicly available, then kindly provide a direct link to the dataset in 'Data Availability'. We noticed that you have already provided a link in 'Describe where the data may be found in full sentences, but that link is a homepage link. Kindly also provide relevant accession numbers or search information in this statement or provide a direct link to the dataset instead of a homepage link. You can also upload a minimal anonymized dataset to the supporting information files.

Authors’ response:Thank you Editor for the comment. As we have mentioned in the data availability section, this study was based on publicly available DHS data. Before the initiation of the study, permission for data access was obtained from major demographic and health surveys through online request from http://www.dhsprogram.com. This data is secondary data that hasn’t any personal identifying information that can be linked to study participants. The confidentiality of data was maintained anonymously. The data used for this study were publicly available with no personal identifier. We have published several research articles using this data and for the details see https://dhsprogram.com/data/dataset_admin/index.cfm.

Response to Reviewers 

Reviewer #1

1. What things did you add for the scientific community or policy makers? Just beyond using advanced statistical technique or what new thing did you find which is not touched by other literatures??

Authors’ response: Thank you reviewer for the comment. As to our search of literature is concerned there are published studies on stunting and associated factors. These studies have treated stunting as a binary outcome (stunted and not stunted). However, stunting has an ordinal nature (not stunted, moderately stunted, and severely stunted) and therefore, collapsing moderately stunted and severely stunted together could result in loss of information as severe stunting and moderately stunted are completely different. Therefore, fitting a model that can consider the ordinal nature of stunting could result in more reliable results and can make valid inferences. Therefore, fitting the multilevel ordinal logistic regression model has both clinical and statistical implications, regarding clinical implications, the occurrence of stunting could not be only affected by individual-level characteristics but also community-level characteristics, this could fully explain the overall variation in the outcome. statistically, given the EDHS data has a hierarchical nature the independence and equal variance assumption are violated. Therefore, to obtain statistically reliable estimates and to make valid inferences, the multilevel ordinal logistic regression model is appropriate. 

2. whereas conducting a multilevel ordinal logistic regression analysis of four models; the null model containing only the response variable, models I and II containing individual and community level variables, respectively, and model III, which contains both individual and community level variables was fitted. Interclass Correlation Coefficient (ICC), Median Odds Ratio (MOR) and

3. Percentage Change in Variance (PCV) was checked to assess the clustering effect. Since these models were nested, we used deviance to check the model comparison and the model with the lowest deviance was chosen as the best model for the data.

4. Why not you usec AIC OR BIC?

Authors’ response: Thank you for the comment. As you can see the four models we have fitted are nested models and therefore for nested models Deviance (-2LLR) is appropriate for model comparison because deviance only considers the model fitness but not the model complexity and therefore it cannot be affected by the number of parameters involved in the model. However, AIC and BIC are measures of model fitness and model complexity, which means they are affected by the number of parameters in the model. That is why we did not use AIC/BIC for model comparison in this study as they are appropriate for unnested models. The statistical analysis and model comparison inform us whether or not to use multilevel analysis based on the assumption (independent observation). The model comparison tells us the comparison between the fitted four models separately. Model 1 (null model) was fitted without independent variables to estimate the cluster-level variation of stunting. Model 2 and Model 3 were adjusted for individual-level variables and community-level variables, respectively. Model 4 was the final model adjusted for individual and community-level variables simultaneously. Finally, we compare those models to identify which model fits the data well (since those four models are nested models deviance is the appropriate model fit index, and the lowest deviance value indicates the best-fitted model). 

5. Over all, you mansucript is well written but you sholud go through to handle grammatical errors throught your document.

Authors’ response: Thank you for the comment. We have extensively edited for typographical and grammatical errors with language experts at the university. 

Reviewer #2

1. Good study. Please elaborate on how the sample size was calculated. Other than that, it is a good, interesting paper. Introduction is precise, methodology is detailed, statistical analysis is appropriate.

Authors’ response: Thank you for the comments. As we have mentioned in the method section, we have used secondary data (DHS) for this study and therefore, our sample size was all under-five children who had anthropometric data. We have included the details in the method section

---

## [Decision Letter · Decision Letter 1]

8 Feb 2024

Prevalence of stunting and associated factors among under-five children in sub-Saharan Africa: Multilevel ordinal logistic regression analysis modeling

PONE-D-23-10610R1

Dear Dr. Seretew,

We’re pleased to inform you that your manuscript has been judged scientifically suitable for publication and will be formally accepted for publication once it meets all outstanding technical requirements.

Kind regards,

Anteneh Mengist Dessie, MPH

Academic Editor

PLOS ONE

Additional Editor Comments (optional):

Reviewers' comments:

Reviewer's Responses to Questions

**Comments to the Author**

1. If the authors have adequately addressed your comments raised in a previous round of review and you feel that this manuscript is now acceptable for publication, you may indicate that here to bypass the “Comments to the Author” section, enter your conflict of interest statement in the “Confidential to Editor” section, and submit your "Accept" recommendation.

Reviewer #1: (No Response)

Reviewer #2: All comments have been addressed

2. Is the manuscript technically sound, and do the data support the conclusions?

Reviewer #1: (No Response)

Reviewer #2: Yes

3. Has the statistical analysis been performed appropriately and rigorously? 

Reviewer #1: (No Response)

Reviewer #2: Yes

4. Have the authors made all data underlying the findings in their manuscript fully available?

Reviewer #1: (No Response)

Reviewer #2: Yes

5. Is the manuscript presented in an intelligible fashion and written in standard English?

Reviewer #1: (No Response)

Reviewer #2: Yes

6. Review Comments to the Author

Reviewer #1: (No Response)

Reviewer #2: The overall issues have been addressed by the authors which is good.

The overall issues have been addressed by the authors which is good.

7. PLOS authors have the option to publish the peer review history of their article (what does this mean?). If published, this will include your full peer review and any attached files.

Reviewer #1: No

Reviewer #2: No

---

## [Editor Report · Acceptance letter]

22 May 2024

PONE-D-23-10610R1 

PLOS ONE

Dear Dr. Seretew, 

I'm pleased to inform you that your manuscript has been deemed suitable for publication in PLOS ONE. Congratulations! Your manuscript is now being handed over to our production team.

Kind regards, 

on behalf of

Mr. Anteneh Mengist Dessie 

Academic Editor

PLOS ONE